# Immune-Deficient Pfp/Rag2^−/−^ Mice Featured Higher Adipose Tissue Mass and Liver Lipid Accumulation with Growing Age than Wildtype C57BL/6N Mice

**DOI:** 10.3390/cells8080775

**Published:** 2019-07-25

**Authors:** Sandra Winkler, Madlen Hempel, Mei-Ju Hsu, Martin Gericke, Hagen Kühne, Sandra Brückner, Silvio Erler, Ralph Burkhardt, Bruno Christ

**Affiliations:** 1Department of Visceral-, Transplantation, Thoracic and Vascular Surgery, Division of Applied Molecular Hepatology, University of Leipzig, 04103 Leipzig, Germany; 2Institute of Anatomy and Cell Biology, Martin-Luther University Halle-Wittenberg, 06108 Halle (Saale), Germany; 3Institute of Biology, Animal Ecology, Martin-Luther-University Halle-Wittenberg, 06099 Halle, Germany; 4Institute of Laboratory Medicine, Clinical Chemistry and Molecular Diagnostics, University of Leipzig, 04103 Leipzig, Germany

**Keywords:** hepatosteatosis, innate immune system, aging, liver, adipose tissue, hepatocyte

## Abstract

Aging is a risk factor for adipose tissue dysfunction, which is associated with inflammatory innate immune mechanisms. Since the adipose tissue/liver axis contributes to hepatosteatosis, we sought to determine age-related adipose tissue dysfunction in the context of the activation of the innate immune system fostering fatty liver phenotypes. Using wildtype and immune-deficient mice, we compared visceral adipose tissue and liver mass as well as hepatic lipid storage in young (ca. 14 weeks) and adult (ca. 30 weeks) mice. Adipocyte size was determined as an indicator of adipocyte function and liver steatosis was quantified by hepatic lipid content. Further, lipid storage was investigated under normal and steatosis-inducing culture conditions in isolated hepatocytes. The physiological age-related increase in body weight was associated with a disproportionate increase in adipose tissue mass in immune-deficient mice, which coincided with higher triglyceride storage in the liver. Lipid storage was similar in isolated hepatocytes from wildtype and immune-deficient mice under normal culture conditions but was significantly higher in immune-deficient than in wildtype hepatocytes under steatosis-inducing culture conditions. Immune-deficient mice also displayed increased inflammatory, adipogenic, and lipogenic markers in serum and adipose tissue. Thus, the age-related increase in body weight coincided with an increase in adipose tissue mass and hepatic steatosis. In association with a (pro-)inflammatory milieu, aging thus promotes hepatosteatosis, especially in immune-deficient mice.

## 1. Introduction

Obesity is one of the most prevalent health problems worldwide, which has been attributed to changes in lifestyle by increasing food intake and/or reducing physical activity. It is mostly associated with metabolic co-morbidities such as diabetes type 2 and non-alcoholic fatty liver diseases (NAFLD), which may progress to chronic inflammatory non-alcoholic steatohepatitis (NASH), cirrhosis and hepatocellular carcinoma (HCC) [1,2]. Epidemiological studies [3,4,5] suggest that the male population carries an increased risk to develop HCC, and the body mass index correlates with the incidence to develop HCC in overweight males, but not in males with normal body weight [6].

Due to its anatomical position in the mesentery, the visceral adipose tissue (VAT) directly communicates with the liver [7]. Non-invasive CT and MR imaging techniques demonstrated that visceral adipose tissue mass correlated with hepatic steatosis associated with obesity and the metabolic syndrome [8,9]. The molecular mechanisms connecting fatty liver pathologies with adipose tissue (dys)function remain largely elusive, yet, hepatic insulin resistance seems one major cause to trigger hepatic lipid storage in obesity. In recent years, it became obvious that a considerable number of obese individuals remain insulin sensitive and lack cardiometabolic complications such as diabetes, dyslipidemia, and vascular disease. These so-called metabolically healthy obese (MHO) are not at higher health risks compared with the normal non-obese population. The underlying mechanisms, which discriminate MHO from metabolically unhealthy obese (MUO) are largely unknown, yet, seem to rely on different immune mechanisms in the context of adipose tissue metabolism at least in part [10,11].

In healthy individuals, the adipose tissue immune state is characterized by an anti-inflammatory milieu, created by eosinophils, alternatively activated macrophages and regulatory T cells secreting anti-inflammatory mediators like IL4, IL13, and IL10. These cytokines are thought to maintain insulin sensitivity and low lipolytic activity of adipocytes as well as to suppress T cell activation and a pro-inflammatory (Th1) response. However, obesity provokes a Th1 response, initiated by hypertrophic adipocytes secreting inflammatory adipokines like leptin, resistin, and angiotensin II as well as generating so-called damage-associated molecular patterns (DAMPs) by necrotic cell death and lipids associated with lipotoxicity. Activation of T cells and conversion of the anti-inflammatory into the inflammatory phenotype of macrophages generate a vicious cycle, by which mainly the inflammatory cytokines TNFα and IFNγ aggravate the Th1 response as well as adipocyte dysfunction evidenced by insulin resistance and increased lipolysis [12,13,14]. It is hence obvious that the crosstalk between liver and inflamed visceral adipose tissue may contribute to the initiation and perpetuation of inflammatory processes in the liver associated with obesity and the metabolic syndrome [15,16,17]. Besides obesity, age- and gender-related differences associated with metabolic deregulation due to mitochondrial dysfunction play an important role in lipid utilization or deposition in both adipose and non-adipose tissues like the liver. These events are not necessarily related to obesity but rather to aging and create a low-grade inflammatory milieu, which increases the susceptibility to metabolic dysfunction in adipose tissue and liver [18,19].

In summary, adipose tissue functional homeostasis is sensed by the liver along the adipose tissue/liver axis. Metabolic dysregulation in the adipose tissue shifts liver lipid utilization into storage provoking lipotoxicity, cell damage and death, and inflammatory tissue remodeling in the liver. In addition, activation of the innate immune system with increasing age has been assigned to cause chronic low-grade inflammation [20,21], further aggravating metabolic dysregulation in adipose tissue and liver. In an immune-deficient (Pfp/Rag2^−/−^) mouse model, we have recently shown that the innate immune system was sufficient to mediate hepatic inflammation in the pathogenesis of NASH [22]. Therefore, we aimed to investigate, whether aging immune-deficient Pfp/Rag2^−/−^ mice lacking B- and T- as well as NK-cells were prone to develop hepatosteatosis under normal feeding conditions, and whether this might be paralleled by changes in visceral adipose tissue mass.

## 2. Materials and Methods

### 2.1. Animal Experiments

All animal experiments were approved by the federal state authority of Saxony (reg.no. TVV15/16) and followed all legislation of the animal welfare act. Immune-deficient Pfp/Rag2^−/−^ (C57BL/6N(B6.129S6-Rag2(tm1Fwa)Prf1(tm1Clrk))) and corresponding wildtype C57BL/6N male and female mice were housed under standard conditions with a 12-hour circadian rhythm at ambient temperature with free access to food (chow diet, V1534, Ssniff, Soest, Germany) and water.

Animals were matched for age, body weight and sex (Table 1). The following abbreviations are subsequently used for the groups (a) male young mice: MY, (b) female young mice: FY, (c) male adult mice: MA, (d) female adult mice: FA, (e) wildtype C57BL/6N: WT and (f) knockout Pfp/Rag2^−/−^: KO (Table 1).

For all experiments, livers and adipose tissue were immediately frozen and stored at −80 °C, cryopreserved for cryosections, or fixed with 4% Histofix. Body, liver, and adipose tissue weight were measured and organ-to-body weight ratios were calculated. Blood samples were taken and serum was stored at −40 °C.

Primary hepatocytes were isolated from young and adult, female and male wildtype and knockout mice (cf. Table 1) after anesthesia with Narcoren^®^, performing a two-step liver perfusion with collagenase (NB4G, Serva Electrophoresis GmbH, Heidelberg, Germany) essentially as described [23,24].

### 2.2. Cell Culture Experiments

Primary hepatocytes, prepared from 3–5 separate animals in each group, were seeded (200.000 cells/mL) in MEM (Merck, Darmstadt, Germany) containing 2% FCS (Gibco, Darmstadt, Germany) and cultured for 3 h. Then the medium was changed to standard hepatocyte growth medium (HGM) [24], to HGM supplemented with 0.5 M palmitic acid, or to HGM without methionine and choline (MCD; HGM w/o methionine and choline, c.c.pro Oberdorla, Germany), each containing EGF (epidermal growth factor, Peprotech GmbH, Hamburg, Germany) and HGF (hepatocyte growth factor, PAN-Biotech GmbH, Aidenbach, Germany) as described previously [24]. Experiments were run for 3 h, 1, 3 or 5 days without medium change. At each point in time, fluorescent detection of lipids was performed as follows. Coverslips with hepatocytes were fixed in 10% formaldehyde for 15 min and incubated with Oil Red O (Sigma-Aldrich Chemie GmbH, Taufkirchen, Germany) for 60 min. After several washing steps with aqua dest, nuclei were stained with Prolong^®^ Gold Antifade Mountant with DAPI (Molecular Probes^TM^, Thermo Fisher Scientific, Dreieich, Germany). Images were taken using the Zeiss Axio Observer.Z1 microscope and lipids quantified by using the Image J 1.48 software (National Institutes of Health, Bethesda, MD, USA).

### 2.3. Serum Analysis

Serum levels of aminotransferases (ALT, AST), triglycerides (TG), cholesterol, free fatty acids (FFA), and glucose of animals (*n* = 4 in each group) were determined in the central laboratory of the University of Leipzig Medical Center on a Cobas (Roche Diagnostics) clinical chemistry analyzer [25].

### 2.4. Histology

Liver cryosections (10 µm) were stained with Sudan III (Carl Roth GmbH + Co. KG, Karlsruhe, Germany) to identify tissue lipids. Nuclei were counterstained with hemalaun (Merck, Darmstadt, Germany). Images were taken using the Zeiss Axio Imager.A1 microscope and the amount of lipids quantified by using the Image J 1.48 software (National Institutes of Health, Bethesda, MD, USA). From each animal (*n* ≥ 6 in each group), 3–6 liver slices were prepared, and 10–15 microscopic fields analyzed.

To assess tissue architecture, paraffin sections (1.5 µm) of the liver and adipose tissue were deparaffinized and stained with hemalaun and eosin (HE, both Merck, Darmstadt, Germany). HE histology of visceral gonadal white adipose tissue (termed VAT in the following) was used to determine the adipocyte size as described previously [26]. Images were taken using an Olympus BX40 epifluorescence microscope.

### 2.5. Immunofluorescence

Paraffin slices were dewaxed, rehydrated, followed by antigen retrieval with Tris-EDTA buffer (pH 9.0) for 30 min, blocking with 5% goat serum (c.c.pro, Oberorla, Germany) for 20 min, and 5% BSA with 0.5% Tween20 for 80 min. For co-staining of E-cadherin (1:200, 610182, BD eBioscience, Heidelberg, Germany) and glutamine synthase (GS) (1:1000, ab16802, abcam, Cambridge, UK), E-cadherin and CYP2E1 (1:200, ab28146, abcam, Cambridge, UK), and E-cadherin and ZO-1 (1:200, 40-2200, Invitrogen, Thermo Fisher, Dreieich, Germany), both primary antibodies were supplied as a mixture overnight. After several washing steps, the first corresponding secondary antibody (1:200, goat anti-mouse, Cy3-labeled, 115-165-003, Dianova, Hamburg, Germany) against E-cadherin was added followed by a washing step. Incubation with the second corresponding secondary antibody (1:200, goat anti-rabbit, AlexaFlour488-labeled, A11008, Life Technologies, Ober-Olm, Germany) was for 50 min. Finally, nuclei were stained with DAPI (1:1000, Carl Roth GmbH + Co. KG, Karlsruhe, Germany) and mounted with 50% glycerol and lacquer. Images were taken using the Zeiss Axio Observer.Z1 microscope. Quantitative analysis was performed including 6–8 animals in each group. 10–15 microscopic fields on one liver slice from each animal were used for image analysis.

### 2.6. Gene Expression Analyses

Total RNA was isolated from 4–8 independent hepatocyte preparations in each group after 3 h, 3 d, and 5 d of cultivation using QIAzol Lysis Reagent (Cat. No. 79306, Qiagen, Hilden, Germany), and the quality of the RNA was monitored by NanoVue Spectrophotometer (4282 V2.0.3, GE Healthcare, Solingen). 2 µg of RNA with OD ratios at 260/280 and 260/230 of 1.621–1.864 and 1.475–2.068, respectively, were used for cDNA synthesis using the Maxima H Minus First Strand cDNA Synthesis Kit (Cat. No. K1652, Thermo Fischer Scientific, Dreieich, Germany) according to the manufacturer’s instructions. Specific gene fragments were amplified by PCR using 0.1 µg of cDNA, PCR Master Mix (2X) (Cat. No. K0171, Thermo Fischer Scientific, Dreieich, Germany), and 0.625 µM of each primer (Table 2). To confirm the size of the fragments and expression levels, PCR products were analyzed on 2% agarose gels in Tris/Borate/EDTA (TBE) buffer, and gel images were acquired by the MicroChemi 4.2 and the ArgusX1 software (Biostep, Burkhardsdorf, Germany). Band intensity was quantified by ImageJ 1.48 and normalized by using the reference genes β₂-microglobulin (B2M), or hypoxanthine-guanine phosphoribosyltransferase (HPRT1).

### 2.7. Adipokine Profiling

The Proteome Profiler^TM^ Mouse Adipokine Array Kit (ARY013, R&D systems, Wiesbaden, Germany,) detecting 38 adipokines (Appendix A) was used to determine the adipokine profile of serum and adipose tissue. 50 µL of serum or 400 µg protein in lysates from adipose tissue were prepared and processed according to the manufacturer’s instructions. For semi-quantitative analyses, labeled proteins were visualized by the MicroChemi 4.2 using the Gel Capture 7.018 software. Pictures with an exposure time of 9 min were quantified using the Image J 1.48 software. Results were corrected for background signals (negative control spots) and normalized to reference spot signals (set equal to 100). To compare results from serum and adipose tissue, relative values were normalized to 1000 µg of total protein. Each group comprised of serum and tissue samples from 4 animals.

### 2.8. Statistics

Data are shown as mean±SEM. Results were analyzed for normal distribution by Shapiro-Wilk testing. For calculation of statistical differences, the Student’s t-test was performed (* *p* ≤ 0.05) unless noted otherwise. For the detection of coherences, the cell culture experiments were analyzed by a general linear model and univariate tests of significance for value. Statistical comparisons for the results of gene expression experiments were made using ordinary 2-way ANOVA with Tukey’s multiple comparison test. Differences between groups were considered significant if the *p*-value was *p* ≤ 0.05 (denoted “*”; *p* ≤ 0.01, ≤ 0.001 and ≤ 0.0001 are denoted as “**”, “***”, and “****”, respectively). Statistics were performed by SPSS 24. Results shown in Figure 7 were obtained using GraphPad Prism 7.

## 3. Results

### 3.1. The Age-Related Increase in Body Weight Coincided with a Disproportionate Increase in Adipose Tissue in Immune-Deficient, but Not Wildtype Mice

The bodyweight of female and male mice (cf. Table 1 for group definition) and the gonadal visceral adipose tissue (VAT), as well as liver mass relative to the total body weight, were determined. With increasing age, wildtype male, but not female animals significantly gained body weight (Figure 1a, left panel). Wildtype young and adult female (WTFY vs. WTFA), as well as young and adult male (WTMY vs. WTMA) mice, displayed a proportionate increase of organ relative to body weight (Figure 1c,e, left panel). Only in female mice, the liver-to-body weight ratio was slightly, albeit significantly, lower in adult vs. young animals (Figure 1c, left panel). Interestingly, no significant increase in liver and VAT weight were observed with increasing age (Figure 1b,d, left panel). Immune-deficient mice of both genders gained body and liver weight with increasing age (KOFY vs. KOFA and KOMY vs. KOMA). The increase in liver mass was proportional to the increase in body weight in both sexes (Figure 1a–c, right panels). However, VAT mass was higher in both, KOMY and KOMA, as compared with the corresponding KOFY and KOFA mice (Figure 1d, right panel). The VAT/body weight ratio was 3-times higher in KOMA than in KOMY mice, and 2-times higher in KOFA than in KOFY animals, respectively. Comparing KOMY and KOFY or KOMA and KOFA, ratios were significantly higher in male than in female animals (Figure 1e, right panel). In summary, the physiological increase in body weight with age coincided with a disproportionate increase in VAT mass in male and to a minor extent in female immune-deficient, but not in wildtype mice.

The metabolic status of the animals was approached by the determination of clinical laboratory serum parameters. Transaminases and total plasma protein content were similar in all groups indicating no apparent liver damage or dysfunction except for WTFA displaying significantly lower plasma protein content as compared with WTFY (54.6 g/dL ± 1.81 vs. 49.1 g/dL ± 1.06; *p* = 0.039). Adult knockout (KOFA and KOMA) mice displayed significantly higher serum glucose levels as compared to young mice (KOFY vs. KOFA: 8.91 mmol/L ± 0.30 vs. 10.91 mmol/L ± 0.35; *p* = 0.011. KOMY vs. KOMA: 6.25 mmol/L ± 0.97 vs. 11.34 mmol/L ± 0.67; *p* = 0.003). This might point to lower glucose tolerance in adult than in young knockout mice. In KOFA mice, cholesterol, triglyceride, and free fatty acid levels were significantly elevated over levels in WTFA mice (Appendix A) suggesting functional differences in lipid metabolism between adult WT and KO mice.

### 3.2. The Age-Related Increase of Visceral Adipose Tissue Mass in Immune-Deficient Mice Was Likely Attributable to Hyperplasia

The increase of adipocyte size may indicate adipose tissue dysfunction due to perturbation of adipocyte lipid metabolism and tissue inflammation [27,28]. Consistent with the disproportionate increase in VAT mass in KOFA and KOMA mice, the diameter of adipocytes and the size distribution were shifted to a higher proportion of larger adipocytes in KOFA and KOMA mice as compared to KOFY and KOMY, resp. (Figure 2b). This shift was also seen in WTMA vs. WTMY and WTFA vs. WTFY (Figure 2a). Thus, later age coincided with an increase in adipocyte diameter in both knockout and wildtype animals. The increase in adipocyte diameter with later age in WTFA and WTMA mice was not accompanied by an increase of VAT mass (cf. Figure 1d,e, left panel). Contrary, the increase in adipocyte diameter in KOFA and KOMA mice was accompanied by a significant increase in VAT weight and VAT/body weight ratio (cf. Figure 1d,e, right panel). This suggests that wildtype adipocytes were likely hypertrophic, whereas immune-deficient mice featured additional VAT hyperplasia.

### 3.3. Adipose Tissue Hypertrophy and Hyperplasia, Respectively, Coincided with Enhanced Fat Storage in Livers of WTFA and KOFA as Well as of KOMA Mice

Adipose tissue dysfunction is closely related to the pathogenesis of fatty liver diseases [29,30]. Therefore, adipose tissue hypertrophy and hyperplasia, primarily in adult mice (WTFA and WTMA; KOFA and KOMA), might impact on liver lipid storage. Lipid accumulation was significantly higher in WTFA than in WTFY mice and reached the highest level over groups (6.06 ± 1.90 %) (Figure 3a). Lipid content was significantly higher in livers of KOMA as compared to KOMY mice, while only a non-significant tendency was evident in the livers of KOFA vs. KOFY mice (Figure 3b). Thus, adipose tissue hypertrophy observed in wildtype mice may mainly promote liver lipid storage in aged female wildtype mice, whereas adipose tissue hyperplasia in knockout animals promoted liver lipid storage with growing age in male, and to a minor extent in female immune-deficient mice.

### 3.4. Livers of WTFA and KOMA Mice Presented Damage-Associated Signs of Tissue Remodeling

We have recently shown that inflammatory steatosis disturbed liver tissue integrity as verified by the loss of E-cadherin and ZO-1 indicating perturbation of cell-cell contacts [31]. Here, we investigated tissue homeostasis by the co-stain and image quantification of E-cadherin (E-cad) representing hepatocytes in the periportal areas of the liver lobule and glutamine synthase (GS) as well as cytochrome P450 subtype 2E1 (CYP2E1) denoting hepatocytes in the perivenous areas (Figure 4). Except for WT male mice, E-cad was significantly lower in adult as compared to young animals. At the same time, GS was higher in WTFA than in WTFY, and in KOMA than in KOMY. The abundance of CYP2E1 was significantly increased in WTFA vs. WTFY, in WTMA vs. WTMY, and in KOMA vs. KOMY. In the liver, lipid metabolism is heterogeneously distributed in the parenchyma, i.e., utilization in the periportal and storage in the perivenous areas [32,33]. Therefore, the enlargement of the perivenous zone, here shown by the increase in the CYP2E1 and GS expressing areas, on the expense of the periportal zone, as shown by the decrease of E-cad expressing areas, indicates a shift from lipid utilization to lipid storage with increasing age. This was most obvious in WTFA and in KOMA mice, thus verifying results as shown in Figure 3.

Fatty liver diseases are associated with epithelial-to-mesenchymal transitions (EMT) indicating the loss of parenchymal organization in connection with tissue remodeling [31,34]. Here, we assessed tissue integrity by immunofluorescent detection of adherens and tight junction markers, E-cad and ZO-1, respectively. Except for KOMA mice, where both cytoplasmic and membranous localization of E-cad was visible, E-cad was predominantly localized at the cell membrane of hepatocytes in WT and KO animals (Figure 5, E-cadherin). ZO-1 is a protein of tight junctions in the canalicular membranes at the apical site of the hepatocytes. This localization was confirmed in WT mice, both female and male (Figure 5a). In KOMA mice, ZO-1 was hardly visible indicating the maceration of cell-cell contacts, and thus the perturbation of parenchymal integrity (Figure 5b).

In summary, these data show the favor of hepatic lipid storage in elder mice, which coincided with functional and morphological changes in the hepatic parenchyma. These were most pronounced in WTFA and in KOMA mice.

### 3.5. The Immune-Deficient Phenotype Favored Lipid Storage in Isolated Hepatocytes

Data so far demonstrated that regulation of hepatic lipid storage is related to age, gender, and the immune phenotype. In order to demonstrate, whether differences depended on immanent features of the hepatocyte, isolated hepatocytes from male/female and young/adult WT and KO mice were cultured for 3 h, 1, 3, and 5 days in hepatocyte growth medium, HGM, and lipid content determined by quantification of lipid droplet areas after staining with Oil Red O. In order to show, whether differences were observable under “normal” or lipid stress conditions, hepatocytes were also challenged by culture in two different steatosis-inducing media, i.e., in a methionine-choline-deficient medium (MCD), or in HGM medium containing 0.5 mM palmitic acid (C16:0). Under each condition tested, hepatocytes accumulated lipid over time. While the rate of accumulation as deduced from the slope of the linear regression line was similar for all groups under ‘normal’ culture conditions, it was increased in all groups under steatosis-inducing conditions. In young female mice, the rate of lipid accumulation was not different between wildtype and knockout animals (Figure 6a, top panels). However, hepatocytes from KOFA animals accumulated roughly twice as much lipid as compared to wildtype animals (Figure 6a, bottom panels). Hepatocytes from both young and adult male immune-deficient animals accumulated lipids to twice as high amounts as compared to wildtype young males (Figure 6b, top panels). Thus, there is a clear correlation between lipid accumulation and the immune-deficient phenotype, which is apparent in hepatocytes from female and male adults, not young mice under “normal” culture conditions. Steatosis-inducing conditions aggravate this correlation in hepatocytes from young and adult males, as well as in hepatocytes from adult female animals.

Taken together, the data show that lipid storage in hepatocytes was fostered by the immune-deficient phenotype primarily in hepatocytes from adult mice, esp. under lipid stress conditions.

### 3.6. Isolated Hepatocytes from KOMA Mice Featured Low Lipogenic and Low Fatty Acid Utilization Capacity

Data from isolated hepatocyte cultures may suggest that differences in lipid accumulation might be due to differences in the lipid metabolism capacity. We assessed general liver function by detection of albumin mRNA, and fatty acid utilization as well as de novo lipogenesis by quantification of PPARα and FABP1, and SREBP1c, ACLY, CPT1, and FASN mRNAs, respectively, at 3 h of culture (Figure 7), and after 3 and 5 days of culture (Appendix A). At 3 h of culture, two major differences became obvious: (1) higher expression of genes in hepatocytes from WTMA vs. KOMA (except for CPT1 and FASN), and (2) higher expression in hepatocytes from KOMY vs. KOMA animals. This indicated a lower capacity of fatty acid utilization and de novo lipogenesis in KO vs. WT mice, and in KOMA vs. KOMY animals, respectively. Thus, immune-deficiency and later age may have favored lipid storage in male adult mice. Since KOMA mice featured high lipid storage in the liver in vivo (cf. Figure 3b), it may be assumed that a defect in utilization is the underlying cause. Contrary, since hepatocytes from WTFA and WTFY animals featured similar expression of lipid metabolism genes, the higher hepatic lipid content in WTFA vs. WTFY (cf. Figure 3a) did not reflect pathological steatosis. This is also underscored by the comparable expression of albumin in isolated hepatocytes of WTFA and WTFY animals. However, albumin was significantly lower in cells from KOMA vs. WTMA, and in KOMA vs. KOMY mice pointing to hepatic impairment in KOMA mice (Figure 7). This corroborated also findings as shown in Figure 4 and Figure 5.

At 3 and 5 days of culture, a decrease of FABP1, SREBP-1, and PPARα expression was observed, which, however, was not significantly different between groups (Appendix A). Therefore, differences in gene expression in hepatocytes may not account for differences in lipid accumulation at these time points, which were roughly twice as high in immune-deficient as compared to wildtype adult hepatocytes under normal culture conditions. Hence, a defective utilization may be assumed, which became most obvious under lipid stress conditions. This is in line with observations that fatty liver diseases may result from mitochondrial dysfunction leading to reduced oxidation under conditions of constant high supply with fatty acids fueling triglyceride storage [35].

### 3.7. Adipose Tissue-Derived Adipokines and Cytokines Potentially Mediate Interactions with the Liver

It has been shown recently that inflammatory markers of visceral adipose tissue may predict fatty liver pathologies in humans [36]. Since there was an obvious difference in hepatic lipid storage between wildtype and immune-deficient elder male mice, we sought for adipokines and cytokines, which might link pathological changes in the adipose tissue to fatty liver phenotypes, in serum and in particular in adipose tissue of KO male adult vs. young mice. Differences were obvious in serum adipokines between WT and KO mice, as well as between KOMY and KOMA animals (Table 3). Age-related differences were found for the abundance of FGF acidic, IGFBP1, and oncostatin M in WT mice, and for AgPR, IGFBP5, pentraxin3, and serpin E1 in KO mice. Age-related higher abundance of ANGPL3, fetuin A, DPPIV, endocan, IGFBP2, IGFBP5, MCSF, pentraxin3, and RBP4 was found across genotypes in KOMY vs. WTMY. ANGPL3, DPPIV, fetuin A, ICAM1, IGF1, IGFBPs (1–3, 6), IL-11, leptin, resistin, and serpinE1 were significantly higher in KOMA vs. WTMA mice. Adiponectin, CRP, IGF1, IGFBP2, IGFBP3, and resistin were detected to levels near or above the upper detection level of the array, primarily in KOMA mice. Even though not significantly different throughout, higher levels of the inflammatory markers CRP, fetuin A, lipocalin2, and of resistin, all markers known to be associated with a fatty liver phenotype, in KO vs. WT mice might point to an adipose tissue-mediated liver steatosis in the immune-deficient mice.

Since the age-related differences in the gain of adipose tissue mass and fatty liver phenotypes were most obvious between KOMA and KOMY mice, we compared VAT derived from KOMY and KOMA mice in regard to adipokines and cytokines indicative of tissue inflammation and/or lipogenic hepatotropism. Factors involved in the regulation of food intake, AgRP (Agouti-related peptide) and leptin, as well as the adipogenic growth factor FGF acidic (FGF1), but also the anti-adipogenic Pref-1 (Preadipocyte factor 1), were more abundant in adipose tissue of KOMA as compared to KOMY mice. Also, the inflammatory cytokines of the IL-6 type family members of pro-inflammatory cytokines, IL-11 and LIF, but as well the anti-inflammatory cytokine IL-10 were found to higher abundance in KOMA than in KOMY adipose tissue. Finally, the monocyte recruiting chemokine RANTES (Regulated on Activation, Normal T Cell Expressed and Secreted) was higher in KOMA than in KOMY adipose tissue (Table 4). The inflammatory markers CRP and lipocalin2, as well as resistin, a cytokine elevated in diabetes type 2, were detected in VAT from male KO mice to abundances in the range of the array´s upper detection level. Together with the fact that these cytokines were also highly abundant in the serum of male KO mice, secretion from the VAT may be assumed.

## 4. Discussion

Here, we showed an increase of the adipose tissue/body weight ratio with increasing age in immune-deficient, not wildtype mice. Together with the increase in adipocyte size with age, it may be concluded that adipose tissue mass expansion in wildtype animals was due to adipocyte hypertrophy, and in immune-deficient animals in addition due to tissue hyperplasia. Presuming communication along the adipose tissue/liver axis, and taking expression of genes involved in liver lipid metabolism into account, hepatic lipid accumulation in the context of hypertrophic adipose tissue may be interpreted to serve for physiological energy homeostasis, yet, in the context of hyperplastic adipose tissue might be related to pathophysiological hepatosteatosis associated with low-grade inflammation. This assumption is corroborated by the pro-inflammatory and pro-adipogenic adipokine profile in serum and adipose tissue primarily in aged male, and to a lower extent in female immune-deficient mice.

### 4.1. Age-Related Mechanisms to Promote Adipose Tissue Pathologies in Adult Mice

Adipose tissue is the main body reservoir for energy substrate storage in the form of triglycerides. These are mobilized during fasting periods by the stimulation of lipolysis to provide free fatty acids for consumptive organs for energy production via oxidation and ATP synthesis. The plasticity of the adipose tissue warrants the adaptation to the organism´s energy situation by changing the size and number of adipocytes and stromal cells to maintain adipose tissue homeostasis under normal feeding conditions. Diet-induced adipocyte hypertrophy and hyperplasia are prominent signs of tissue remodeling and are regarded as a compensatory response to nutritional lipid overload exceeding lipid utilization [37,38]. Characteristics of adipose tissue remodeling comprise immune cell infiltration and inflammation, changes in adipokine secretion, and eventually metabolic dysregulation concomitant with systemic low-grade inflammation [39,40,41,42]. In line with the larger diameter and size distribution of adipocytes in adult as compared to young mice together with the increase in the VAT/body weight ratio, an age-dependent hyperplasia of the VAT in the KO mice, not in the WT mice may be assumed. Since in WT mice the increase in diameter and size distribution of adipocytes were observed without an increase in VAT/body weight ratio, aging in WT mice may be associated with VAT hypertrophy. Aging entails the same mechanisms of adipose tissue inflammation, changes in secretory profiles and systemic inflammation as adipose tissue remodeling induced by obesity [43,44]. Therefore, our data support that physiological aging is associated with adipose tissue hypertrophy in WT mice, and that immune-deficiency promotes adipose tissue hyperplasia in addition to hypertrophy.

In the immune-deficient mice, the increase in adipose tissue mass was much more pronounced in male as compared to female adult mice. This is in line with findings in female patients featuring more frequent occurrence of brown adipose tissue (BAT) than male patients, which inversely correlated with the body mass index suggesting a prominent role of BAT in whole-body energy homeostasis [45,46,47]. Indeed, β-adrenergic induction of BAT was more sensitive in female mice as compared to males indicating sex-dependent recruitment of BAT in visceral adipose tissue, which might favor a protective role in females by utilizing stored lipids for thermogenesis [48,49].

### 4.2. Hepatic Mechanisms Involved in Increased Lipid Storage in Old Mice

Except for male WT mice, both WT and KO mice featured an increase in hepatic lipid storage with increasing age (cf. Figure 3). Isolated hepatocytes from all groups, however, stored lipids to similar amounts under ‘normal’ culture conditions (cf. Figure 6). In addition, expression of genes involved in both lipid metabolism and liver functions were similar in hepatocytes from both young and adult wildtype mice. Also, no differences were obvious between WT and KO female hepatocytes and WT and KO young male hepatocytes. However, at the beginning of culture, expression of genes involved in lipid utilization was lower in hepatocytes from adult male immune-deficient mice. This might indicate increased lipid storage in these animals by inhibition of FFA utilization as the consequence of the excess provision of FFA from the VAT (cf. Figure 7). At later time-points (3 d and 5 d of culture), expression of genes was similar throughout groups. Therefore, higher lipid accumulation in hepatocytes from KO mice as compared to WT hepatocytes esp. under lipogenic culture conditions must be due to a shift from utilization to storage. This was most obvious in cells from adult mice, which is in line with reports that aging fostered hepatic steatosis *per se* [50,51].

In humans, inflammation of VAT correlated with steatohepatitis [52,53], albeit the mechanisms involved remain largely elusive. In addition, it has been shown that adipose tissue-derived inflammatory markers may predict fatty liver phenotypes in humans [36]. Therefore, VAT-derived mediators, reaching the liver via the bloodstream, might induce or augment inflammatory processes also in the liver. ANGPL3, DPPIV, IGF1 and most of the IGF-binding proteins, all known to be secreted from the adipose tissue in relation to obesity and to pathological lipoprotein and lipid metabolism [54,55,56,57], were elevated in KO vs. WT mice. Also, resistin and serpinE1, derived from adipose tissue to high levels during adipogenesis [58,59,60,61], as well as fetuin A and RBP4 indicative for a fatty liver phenotype [62], were significantly more abundant in KO vs. WT mice. Yet, a decrease in SerpinE expression was among the strongest predictors for fatty liver disease in humans [36]. Therefore, a decrease in SerpinE in adult KO mice, which is not evident in WT mice, can be discussed as one mediator in the adipose tissue-liver crosstalk. Also, the increase in resistin in adult KO mice could contribute to hepatic insulin resistance and increased gluconeogenesis leading to fatty liver disease in adult KO mice [63]. Not surprisingly, pentraxins, promoters of innate immunity [64], were higher in KO vs. WT mice. Thus, adipose tissue-linked systemic inflammation in KO vs. WT mice might be indicated by higher levels of leptin and resistin, adipokines contributing to the systemic low-grade inflammatory milieu in obesity and associated metabolic disorders [65]. Yet, changes in serum cytokines may not necessarily be due to secretion from adipose tissue, but may also derive from other sources like immune cells. Since many of those shown in Table 3 were significantly altered in KO vs. WT mice, it may be assumed that the cytokine milieu in the immune-deficient animals could be linked to pathological changes in adipose tissue and liver.

In the liver parenchyma, metabolic functions are distributed heterogeneously [66]. Hepatocytes surrounding the branches of the central vein exhibit a higher rate of lipogenesis as well as triglyceride and VLDL synthesis as compared to hepatocytes surrounding the branches of the portal vein [32,33,66,67]. This pattern is dynamic and undergoes spatial changes according to nutritional behavior, circadian rhythmicity, and regulation by hormones and innervation [68]. In the study presented, the perivenous parenchyma was significantly enlarged on the expense of periportal areas in adult as compared to young mice. Similarly, the perivenous expression of CYP2E1 expanded more in adult than in young mice (cf. Figure 4). In obese rats, CYP2E1 was induced by overfeeding indicating a functional link between enzyme expression and obesity [69]. The enzyme is involved in ethanol-induced changes in hepatic lipid metabolism [70], and a major contributor of reactive oxygen species [71,72] causing lipid peroxidation in the context of NASH [73,74]. NASH, induced by feeding a methionine-choline-deficient diet in immune-deficient mice, caused disruption of adherens junctions in the liver [31]. In the study presented, both adherens and tight junctions were perturbed without feeding a NASH-inducing diet (cf. Figure 5). Together with the expansion of perivenous areas of the hepatic parenchyma on the expense of periportal areas, this indicated an increased lipid storage potential under normal feeding conditions. This was more obvious in adult than in young, and in male than in female immune-deficient mice denoting aging and immune system dysfunction as risk factors for hepatosteatosis.

## 5. Conclusions/Hypothesis

Adipose tissue hypertrophy and hyperplasia are signs of tissue remodeling associated with lipid metabolism dysfunction in the context of metabolic diseases. Changes in the adipose tissue adipokine/cytokine secretory profile create a local and systemic low-grade inflammatory milieu. This is communicated to the liver either directly along the adipose tissue/liver axis, or indirectly via systemic interactions with other cytokine sources like immune cells, thus promoting fatty liver phenotypes. Since in female mice lipid utilization is supported by energy expenditure in brown adipose tissue under the control of estrogens [48,75], particularly aged male mice are affected even under normal feeding conditions. Gender-related differences in hepatic metabolism and the pathogenesis of non-alcoholic steatohepatitis have also been predicted from computational analyses using a novel tool termed “LiverSex Computational Model” [76]. In line with reports on the disruption of circadian metabolic regulation by dietary fat intake [77], which also affects isolated hepatocytes [78], this conclusion is corroborated by findings that circadian metabolic gene expression in the liver changed with age [79]. This highlights the mutual interference of age- and nutrition-related regulation of hepatic energy metabolism. The emergence of fatty liver phenotypes with growing age may be supported by the aging immune system, which is characterized by the down-regulation of adaptive immune mechanisms and activation of the innate immune system [20,21]. This is in line with the role of adipose tissue-derived adipokines/cytokines in the pathogenesis of systemic low-grade inflammation involving cells of both the adaptive and the innate immune system showing a Th2 response under lean, and a Th1 immune response milieu under obese conditions in the adipose tissue [80,81]. Similar mechanisms enrolling activation and recruitment of inflammatory cells apply to the pathogenesis of fatty liver diseases like NASH [82,83,84].

Hence, our data might indicate that aging predisposes to a shift from hepatic lipid utilization to storage, which might be aggravated by aging immune mechanisms.

## Figures and Tables

**Figure 1 cells-08-00775-f001:**
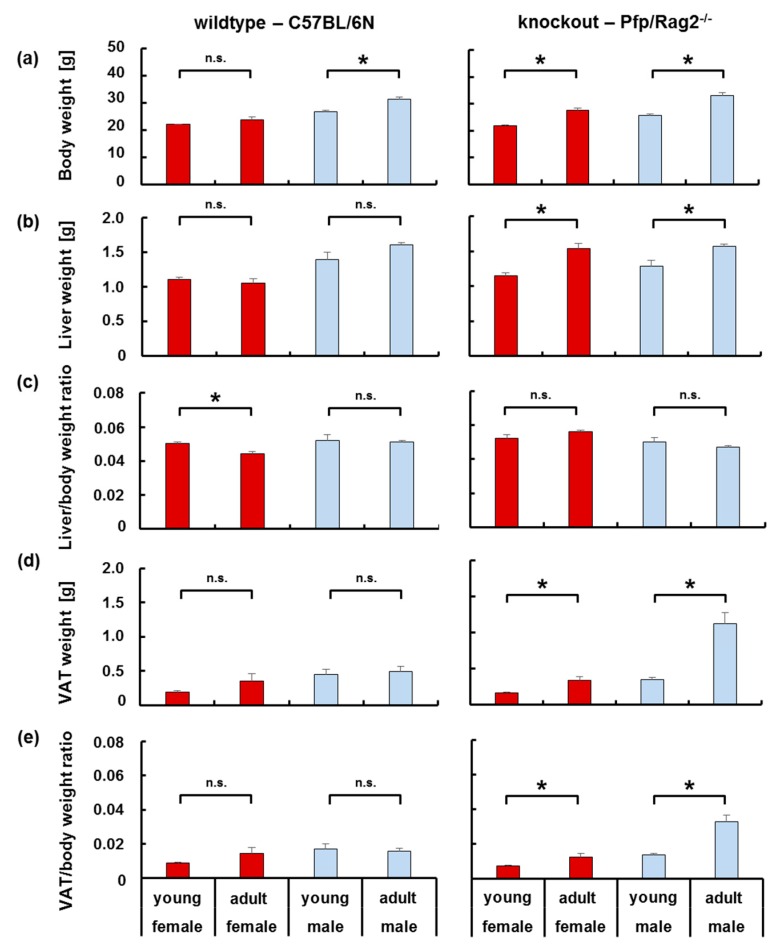
Increase of body, liver and visceral adipose tissue (VAT) weight in wildtype and knockout female and male mice. The proportionate increase in organ weight (**b**,**d**) relative to the increase in body weight (**a**) is given as the organ-to-body weight ratio (**c**,**e**). Data from female and male mice are shown in red (bold shade) and blue (light shade), resp. Horizontal lines marked with an asterisk indicate significant differences at the *p* ≤ 0.05 level; n.s. indicates no significant differences.

**Figure 2 cells-08-00775-f002:**
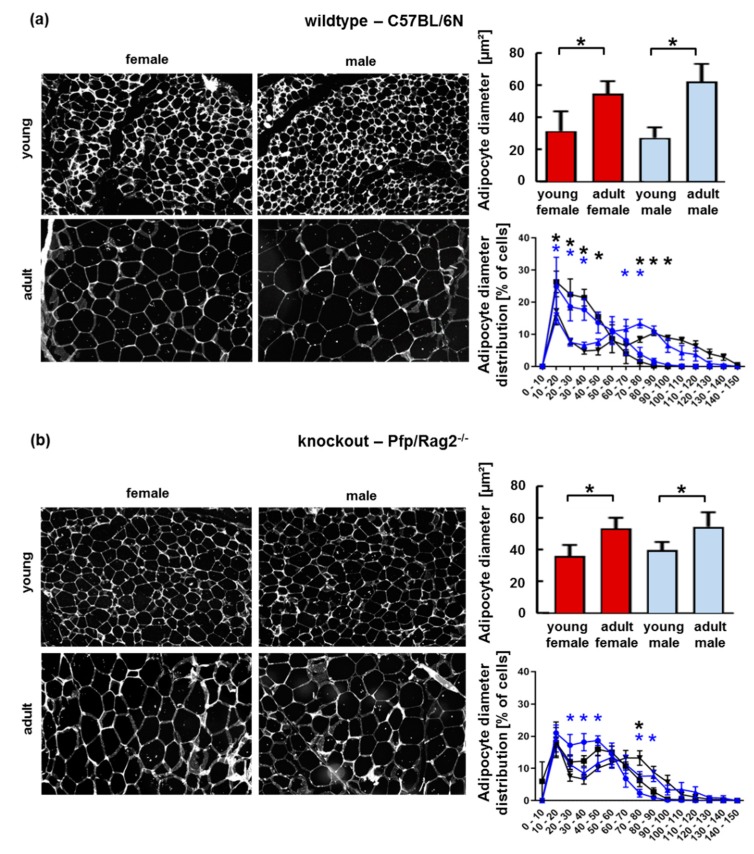
Features of VAT from wildtype and knockout female and male mice. HE staining of adipose tissue revealed larger adipocytes in adult as compared to young animals (**a**,**b**—left panels; shown as binary images (b/w) as used for quantitative analysis). Adipocyte size was quantified by measuring the average diameter of cells from female (red, bold shade) and male (blue, light shade) mice (right: upper panels). Size distribution is shown in the lower right panels. Female mice: young—dots, adult—upward triangles. Male mice: young—squares, adult—downward triangles. Light (blue, female) or bold (black, male) asterisks, resp., indicate significant differences at the *p* ≤ 0.05 level. Histological pictures in (**a**) and (**b**) are representative of 3 different animals in each group.

**Figure 3 cells-08-00775-f003:**
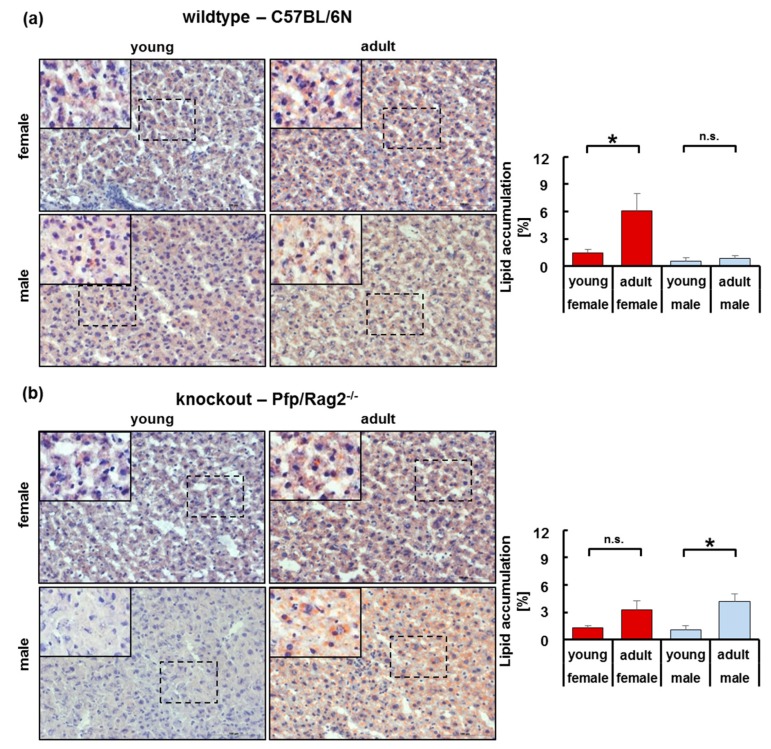
Lipid content in livers from wildtype (**a**) and knockout (**b**) female and male mice. Liver lipids were stained with Sudan III (orange staining, left panels) and quantified by image analysis (right panels, female in bold (red) and male (blue) in light shades). Horizontal lines marked with an asterisk indicate significant differences at the *p* ≤ 0.05 level; n.s. indicates no significant differences. Insets (bold lines) in the histological images show higher magnifications of the areas indicated by dashed lines.

**Figure 4 cells-08-00775-f004:**
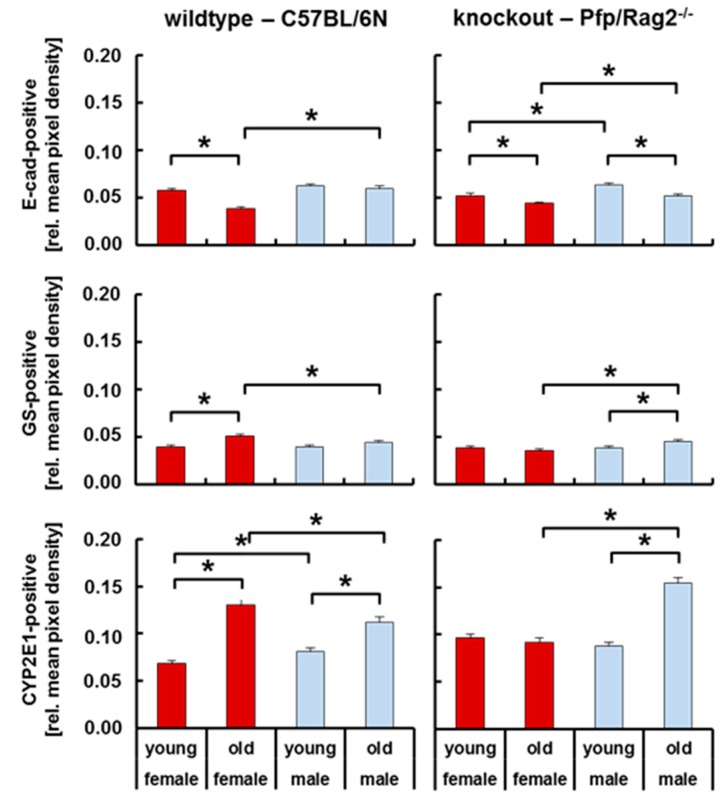
Quantitative expression of periportal E-cadherin (E-cad), and of perivenous glutamine synthase (GS) and Cytochrome P450 2E1 (CYP2E1) in livers from wildtype and knockout female and male mice. E-cadherin, GS, and CYP2E1 abundances were quantified by image analysis of immunofluorescent co-stained E-cadherin and GS as well as E-cadherin and CYP2E1 (Appendix A). The left panels show data from wildtype female (red, bold) and male (blue, light) mice, and the right panels from knockout female (red, bold) and male (blue, light) mice. Horizontal lines marked with an asterisk indicate significant differences at the *p* ≤ 0.05 level.

**Figure 5 cells-08-00775-f005:**
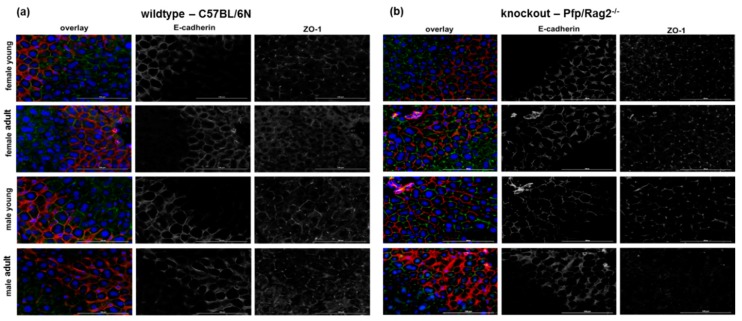
Expression of E-cadherin and ZO-1 in livers from wildtype (**a**) and knockout (**b**) female and male mice. The adherens junction protein E-cadherin (red) and the tight junction protein ZO-1 (green) were detected by immunofluorescence microscopy of liver sections. Pictures on the left side show the overlay, pictures in the middle the E-cadherin stain and right, the ZO-1 stain. Pictures are representative out of 3 different animals in each group.

**Figure 6 cells-08-00775-f006:**
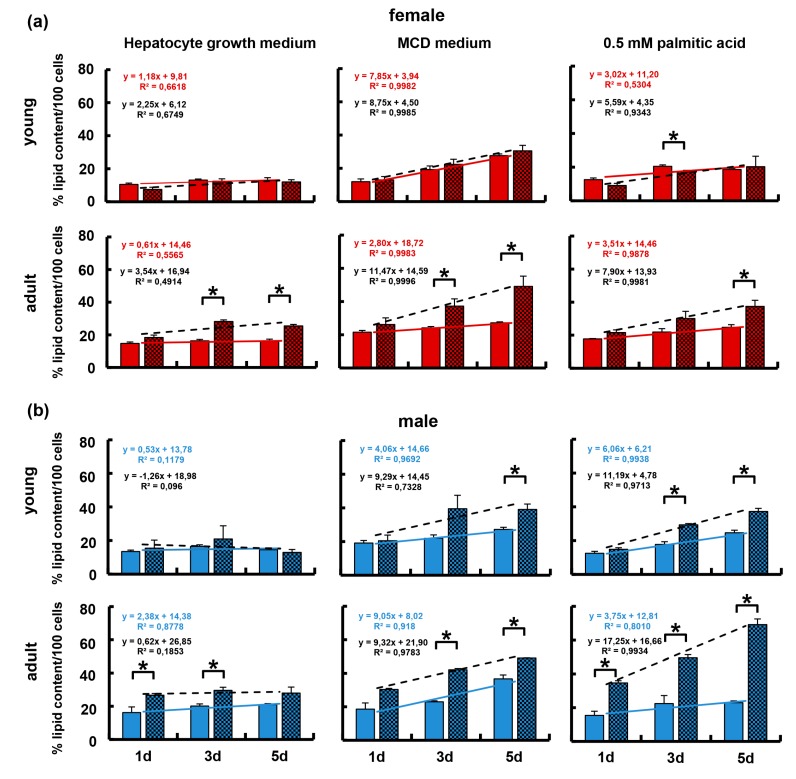
Lipid accumulation by isolated hepatocytes from wildtype and knockout, female and male mice. Hepatocytes were cultured either in hepatocyte growth medium HGM (left panels), or steatosis-inducing MCD medium (middle panels), or in HGM supplemented with 0.5 mM palmitic acid (right panels). In (**a**), wildtype female young and adult mice are shown as solid columns with solid trend lines, equations and coefficients of determination (upper values); knockout female young and adult mice are shown as hatched columns with hatched trend lines, equations, and coefficients of determination (lower values). In (**b**), wildtype male young and adult mice are shown as solid columns with solid trend lines, equations, and coefficients of determination (upper values); knockout male young and adult mice are shown as hatched columns with hatched trend lines, equations, and coefficients of determination (lower values). Hepatocytes were cultured for the times indicated and lipids stained with Oil Red O. Quantification was performed by image analysis and values normalized as the percentage amount of stain per 100 cells. Horizontal lines marked with an asterisk indicate significant differences at the *p* ≤ 0.05 level.

**Figure 7 cells-08-00775-f007:**
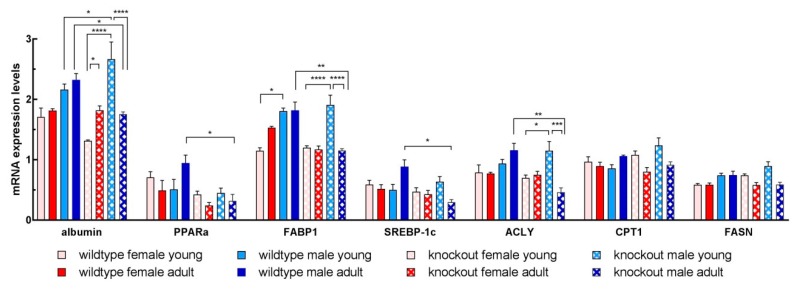
Levels of mRNA coding for proteins involved in lipid metabolism and function in isolated hepatocytes of wildtype and knockout female and male mice Expression was quantified by semi-quantitative analysis of PCR products and the signals were normalized to the reference genes. Differences between groups were considered significant, if the *p*-value was ≤0.05 (denoted “*”; whereas *p* ≤ 0.01, 0.001, and 0.0001 are denoted as “**”, “***”, and “****”, respectively).

**Table 1 cells-08-00775-t001:** Overview of animal biometric parameters.

	Wildtype		Knockout	
	Female	Male	Female	Male
Young	13.7 ± 0.47 weeks	14.3 ± 0.48 weeks	13.8 ± 0.62 weeks	13.6 ± 0.52 weeks
Mean weight [g]	22.11 ± 0.18	26.77 ± 0.55	21.94 ± 0.27	25.64 ± 0.47
Group abbreviation	WTFY	WTMY	KOFY	KOMY
Adult	29.3 ± 0.49 weeks	29.0 ± 1.26 weeks	29.8 ± 1.47 weeks	30.3 ± 0.77 weeks
Mean weight [g]	23.81 ± 1.13	31.44 ± 0.69	27.44 ± 0.98	32.93 ± 1.16
Group abbreviation	WTFA	WTMA	KOFA	KOMA

**Table 2 cells-08-00775-t002:** List of primers used for semi-quantitative RT-PCR analysis.

Gene	Accession No.	Forward Primer 5′ → 3′	Reverse Primer 5′ → 3′
ACLY	NM_001199296.1	CCCAAGTCCAAGATCCCTGC	CTGCTTGTGATCCCCAGTGA
Albumin	NM_009654.4	CCAATCCTCCCGCATGCTAC	GCGAACTAGAATGGCATTTTGGAA
B2M	NM_009735.3	TCTACTGGGATCGAGACATGTGA	ATTGCTATTTCTTTCTGCGTGCAT
CPT1A	NM_013495.2	CCATGATGGACCCCACAACA	ATGGCTCAGACAGTACCTCCT
FABP1	NM_017399.4	GGAAAAAGTCAAGGCAGTCGTC	CTCTTGTAGACAATGTCGCCC
FASN	NM_007988.3	TGCACCTCACAGGCATCAAT	GTCCCACTTGATGTGAGGGG
HPRT1	NM_013556.2	ACAGGCCAGACTTTGTTGGAT	TGCAGATTCAACTTGCGCTC
PPARα	NM_011144.6	GTTCACGCATGTGAAGGCTG	AGCGAATTGCATTGTGTGACAT
SREBP1c	NM_011480.4	GATTTGGCCCGGGGAGATTT	TGGCGGATGAGGTTCCAAAG

**Table 3 cells-08-00775-t003:** Serum adipokines and cytokines.

	Wildtype C57BL/6N				Knockout Pfp/Rag2^−/−^				Significance
	Young (WTMY)		Adult (WTMA)		Young (KOMY)		Adult (KOMA)		
	Relative Mean Pixel Density/mg Serum Protein								
	Mean	SEM	Mean	SEM	Mean	SEM	Mean	SEM	*P*
**Adiponectin**	50.10	0.38	51.20	0.26	51.67	1.07	51.05	0.14	
**AgRP**	1.82	0.89	0.76	0.28	2.53	0.75	0.47	0.24	KOMY/KOMA 0.004
**ANGPL3**	11.90	1.62	16.10	0.50	21.51	6.80	23.82	1.30	WTMY/KOMY 0.022 WTMA/KOMA 0.001
**CRP**	38.79	3.76	44.95	1.61	44.08	4.82	48.81	0.83	
**DPPIV**	11.86	2.16	11.08	0.61	16.59	5.47	21.62	1.58	WTMY/KOMY 0,043 WTMA/KOMA 0.001
**Endocan**	2.34	0.27	2.55	0.15	3.35	1.01	2.83	0.42	WTMY/KOMY 0.033
**FetuinA**	22.87	2.52	18.28	1.17	31.95	7.29	25.19	1.25	WTMY/KOMY 0,034 WTMA/KOMA 0.007
**FGFacidic**	3.28	0.58	1.22	0.12	4.82	2.55	0.93	0.22	WTMY/WTMA 0.013
**ICAM1**	14.53	1.86	10.26	1.04	19.46	6.80	18.31	0.37	WTMA/KOMA 0.000
**IGFI**	37.22	5.83	21.46	5.40	40.62	9.08	49.75	0.23	WTMA/KOMA 0.013
**IGFII**	4.02	0.92	3.98	0.15	5.41	1.08	4.96	0.59	
**IGFBP1**	8.31	2.25	32.31	6.15	12.05	4.98	9.02	1.83	WTMY/WTMA 0.011 WTMA/KOMA 0.011
**IGFBP2**	29.03	3.56	35.86	0.82	36.38	10.66	48.53	0.73	WTMY/KOMY 0.015 WTMA/KOMA 0.000
**IGFBP3**	43.01	2.98	47.01	0.85	42.05	6.59	50.82	0.23	WTMA/KOMA 0.005
**IGFBP5**	4.47	0.57	5.13	0.22	8.83	3.02	6.34	0.65	WTMY/KOMY 0.014 KOMY/KOMA 0.044
**IGFBP6**	18.08	3.13	20.75	2.53	23.59	9.55	37.90	4.73	WTMA/KOMA 0.019
**IL11**	1.14	0.10	1.59	0.27	2.95	1.21	0.68	0.17	WTMA/KOMA 0.030
**Leptin**	1.73	0.21	1.17	0.26	2.83	1.21	2.72	0.53	WTMA/KOMA 0.039
**Liopcalin2**	15.96	1.33	18.60	2.11	29.45	9.48	19.83	1.58	
**MCSF**	3.35	0.89	5.60	0.87	5.69	1.85	6.16	0.59	WTMY/KOMY 0.037
**OncostatinM**	0.82	0.16	1.64	0.22	3.02	1.01	1.14	0.08	WTMY/WTMA 0.023
**Pentraxin2**	5.29	1.94	9.56	1.82	16.90	7.17	5.40	1.02	
**Pentraxin3**	2.13	0.42	2.37	0.34	4.32	1.63	2.11	0.24	WTMY/KOMY 0.038 KOMY/KOMA 0.031
**RBP4**	12.34	1.14	10.30	0.98	16.72	4.68	12.37	1.01	WTMY/KOMY 0.047
**Resistin**	25.96	4.31	24.51	3.03	28.58	11.08	48.34	1.17	WTMA/KOMA 0.000
**SerpinE1**	1.75	0.31	2.06	0.39	3.11	1.15	0.85	0.15	WTMA/KOMA 0.028 KOMY/KOMA 0.017
**TIMP1**	0.52	0.10	0.93	0.20	2.71	0.99	0.50	0.11	

The abundance in serum derived from wildtype male young (WTMY) and adult (WTMA) mice, as well as from knockout male young (KOMY) and adult (KOMA) mice was quantified by the use of the Proteome ProfilerTM Mouse Adipokine Array Kit as described. Only proteins with a relative mean pixel density above 2.5 as lowest confident detection level are shown. The upper detection limit in the serum was at 50 relative mean pixel density.

**Table 4 cells-08-00775-t004:** Adipokines and cytokines in VAT.

	Knockout Pfp/Rag2^−/−^				Significance
	Young (KOMY)		Adult (KOMA)		
	Relative Mean Pixel Density/mg Protein				
	Mean	SEM	Mean	SEM	*p*
**Adiponectin**	249.76	1.05	247.81	3.50	
**AgRP**	8.44	1.35	12.52	0.88	
**ANGPL3**	64.21	3.45	72.86	11.18	
**CRP**	249.80	0.27	249.49	1.34	
**DPPIV**	133.98	9.61	139.65	10.08	
**Endocan**	38.38	8.63	63.27	8.99	
**FetuinA**	183.56	11.24	196.04	6.99	
**FGFacidic**	250.70	0.85	249.87	0.33	
**FGF21**	14.39	2.75	24.10	1.69	0.024
**HGF**	15.26	4.69	26.14	3.40	
**ICAM1**	225.11	8.89	238.95	4.28	
**IGFI**	23.40	3.83	33.52	6.00	
**IGFII**	14.10	0.89	17.69	2.16	
**IGFBP1**	63.02	18.20	31.54	5.64	
**IGFBP2**	135.32	5.14	161.04	23.43	
**IGFBP3**	211.48	6.13	230.40	7.45	
**IGFBP5**	60.57	6.19	82.33	1.99	0.015
**IGFBP6**	241.15	5.05	249.88	0.67	
**IL6**	9.57	1.81	13.54	1.48	
**IL10**	14.67	1.88	23.60	2.49	0.029
**IL11**	15.93	2.44	22.92	1.46	0.049
**Leptin**	141.31	27.71	225.14	7.41	0.027
**LIF**	11.18	1.58	20.27	3.36	0.050
**Liopcalin2**	249.91	1.90	242.53	4.43	
**MCP1**	23.25	4.65	43.67	8.51	
**MCSF**	47.88	6.20	63.50	2.23	0.056
**OncostatinM**	14.31	1.64	16.99	2.05	
**Pentraxin2**	23.80	2.94	19.12	2.71	
**Pentraxin3**	32.22	3.04	43.15	4.21	
**Pref1**	14.04	2.54	24.49	2.46	0.025
**RAGE**	19.37	3.23	21.24	2.43	
**RANTES**	17.97	2.22	32.60	5.21	0.041
**RBP4**	94.19	9.55	92.23	4.47	
**Resistin**	248.49	1.82	245.53	1.03	
**SerpinE1**	60.81	14.27	88.45	26.72	
**TIMP1**	28.02	3.93	47.08	11.84	
**TNFa**	8.52	1.11	12.33	1.59	
**VEGF**	55.13	14.65	55.56	2.77	

The abundance in VAT derived from knockout male young (KOMY) and adult mice (KOMA) was quantified as described. Proteins with a relative mean pixel density above 12.5 as lowest detection level are shown. The upper detection limit in VAT was at 250 relative mean pixel density.

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
