# Peer review of "Immune-Deficient Pfp/Rag2−/− Mice Featured Higher Adipose Tissue Mass and Liver Lipid Accumulation with Growing Age than Wildtype C57BL/6N Mice"

_cells, 2019, doi:10.3390/cells8080775_

Round 1

Reviewer 1 Report

This study investigated the impact of Pfp/Rag2 deletion on visceral adipose tissue and liver mass as well as hepatic lipid storage in young and old mice. The authors found that the physiological age-related increase in body weight was associated with a disproportionate increase in adipose tissue mass in the Pfp/Rag2 KO, which coincided with higher triglyceride storage in the liver. Inflammatory, adipogenic and lipogenic markers were higher in serum and adipose tissue of the Pfp/Rag2 KO mice than in WT mice. The study suggest that a pro-inflammatory milieu in the Pfp/Rag2 deletion mice accounts for aging-provoked hepatosteatosis. Below are comments for revision.

1. Generally speaking, lipid overload in the adipocyte is a causal factor of adipocyte death which , in turn, causes inflammation and ectopic lipid deposition such as steatosis. In the present study, however, the old Pfp/Rag2 KO mice actually exhibited smaller adipocyte size compared to the old WT mice. How do the authors explain the mechanisms underlying the ectopic lipid deposition in the liver of the old Pfp/Rag2 KO mice?

2. The old Pfp/Rag2 KO mice showed remarkable larger VAT mass as well as larger liver weight compared to that of the old WT mice, suggesting a higher energy storage. Did the authors measured the food intake? Is there any difference between the old Pfp/Rag/Rag2 KO mice and old WT mice?

3. The intestine possesses a large number of lymphocytes. Do the authors believe that if the intestinal dysfunction in B cells/T cells may contributes to the phenomena observed in this study.

4. The authors determined lipid storage function of the liver using a 5-day primary hepatocyte culture. It is well known that long term primary hepatocyte culture leads to alternation in expression  of a large number genes. Thus, the gene expression of lipid metabolism-related genes such as fatty acid b-oxidation (Cpt1a, Vcad, Acox1), triglyceride synthesis (Fasn, Acc, Dgat1, Dgat2) and regulators (PPara, Pparg, Srebp1c) should be measured. At the same time, hepatic triglyceride concentrations should be quantitatively measured using biochemical assay method. 

Author Response

We thank this reviewer for his valuable advice and tried to meet all the concerns raised as far as possible by introducing appropriate changes in the text and the presentation of results. This improved the manuscript significantly.

Reviewer 1

This study investigated the impact of Pfp/Rag2 deletion on visceral adipose tissue and liver mass as well as hepatic lipid storage in young and old mice. The authors found that the physiological age-related increase in body weight was associated with a disproportionate increase in adipose tissue mass in the Pfp/Rag2 KO, which coincided with higher triglyceride storage in the liver. Inflammatory, adipogenic and lipogenic markers were higher in serum and adipose tissue of the Pfp/Rag2 KO mice than in WT mice. The study suggest that a pro-inflammatory milieu in the Pfp/Rag2 deletion mice accounts for aging-provoked hepatosteatosis. Below are comments for revision.

1. Generally speaking, lipid overload in the adipocyte is a causal factor of adipocyte death which , in turn, causes inflammation and ectopic lipid deposition such as steatosis. In the present study, however, the old Pfp/Rag2 KO mice actually exhibited smaller adipocyte size compared to the old WT mice. How do the authors explain the mechanisms underlying the ectopic lipid deposition in the liver of the old Pfp/Rag2 KO mice?

The difference between adipocyte size in adult WT and KO is not significant indicating that ageing generally might be associated with an increase in adipocyte size. However, while WT animals did not feature an increase in VAT mass with age, VAT mass increased in both female and male KO mice with age (Figure 1). Therefore, we concluded that the increase in adipocyte size in WT mice was due to hypertrophy in the context of physiological energy homeostasis regulation, while in KO mice the increase in adipocyte size and mass was due to hypertrophy and hyperplasia likely in the context of pathophysiological lipid storage due to perturbation of adipocyte energy homeostasis. This context is outlined in the manuscript under 3.2. and 3.3. Concerning the mechanisms involved in increased hepatic lipid storage in the context of adipose tissue hyperplasia, we may only speculate upon. Since isolated hepatocytes accumulated lipids at comparable rates under non-steatosis inducing conditions (Figure 6), and initial expression of lipid metabolism genes was similar under all conditions, except for KO male adult mice (Figure 7), we assumed that systemic regulation must be involved. This may comprise the increased provision of free fatty acids from the adipose tissue, which is known to foster lipid storage in the liver. This is outlined in the discussion under 4.2. Though there is an increased incidence of fatty liver phenotypes with age, the underlying mechanisms are largely unknown, but may involve autophagy impairment, oxidative stress, or cellular senescence associated with mitochondrial dysfunction (cf. refs. 48,49). We would not like to extend this discussion, because it would even increase speculative statements, which we were advised by reviewer 2 rather to avoid.

2. The old Pfp/Rag2 KO mice showed remarkable larger VAT mass as well as larger liver weight compared to that of the old WT mice, suggesting a higher energy storage. Did the authors measured the food intake? Is there any difference between the old Pfp/Rag/Rag2 KO mice and old WT mice?

Thank you for this valuable hint, and we appreciate that the reviewer comes to the same suggestion, i.e. that the elder KO mice featured obvious higher energy substrate storage than the WT mice. We agree that we should have measured food intake and physical activity of the mice systematically. This would have required special animal housing in metabolic cages, which we have not available in our facilities. We made some inspections of food intake (by weighing remaining food), though not systematically, which revealed no obvious differences between WT and KO animals. This was confirmed by observations of the animal keepers. They also confirmed that there was no obvious difference in the physical activity. However, our initial observation with the KO mice was that they had much more adipose tissue mass than the age-matched WT animals, very obvious independent of food intake. Since it is known that adipose tissue mass correlates with fatty liver phenotypes, and incidence increases with age, we reasoned that the immune-deficient genotype may foster hepatic lipid storage, which we confirmed with this study.

3. The intestine possesses a large number of lymphocytes. Do the authors believe that if the intestinal dysfunction in B cells/T cells may contributes to the phenomena observed in this study.

We have not looked at this point and could only speculate upon. However, it is very likely that immunodeficiency impacts on absorption of nutrients in the gut. A systems biology approach revealed that the absence of B cells caused lipid malabsorption and the decrease of body fat depots (Nat Med. 2011;17(12):1585-93. Crosstalk between B lymphocytes, microbiota and the intestinal epithelium governs immunity versus metabolism in the gut. Shulzhenko N et al.). This is in line with the observation that patients suffering from HIV infections display impairment of intestinal lipid absorption, decrease of serum lipids, and loss of body fat. This is contrary to our observations corroborating that intestinal absorption may not be involved.

4. The authors determined lipid storage function of the liver using a 5-day primary hepatocyte culture. It is well known that long term primary hepatocyte culture leads to alternation in expression  of a large number genes. Thus, the gene expression of lipid metabolism-related genes such as fatty acid b-oxidation (Cpt1a, Vcad, Acox1), triglyceride synthesis (Fasn, Acc, Dgat1, Dgat2) and regulators (PPara, Pparg, Srebp1c) should be measured. At the same time, hepatic triglyceride concentrations should be quantitatively measured using biochemical assay method. 

Completely right! We were quite aware of this fact and determined gene expression profiles in addition to 3h of culture also at 3 and 5 days (results now in the Supplementary Materials, Figure S3). While most of the transcripts did not show remarkable decreases, expression of FABP1 and SREBP-1c were reduced in all groups. PPARa was reversed from higher expression in WT to higher expression in KO animals with ongoing culture. These differences were not significant between groups. Therefore, we do not believe that differences in gene expression in hepatocytes may account for differences in liver lipid content in vivo, which is also reflected by the similar rates of lipid deposition in hepatocytes under normal culture conditions. Differences between WT and KO animals became most obvious under lipid stress conditions, and were most pronounced in adult male KO mice (cf. Figure 6). This is in line with observations that fatty liver diseases may result from mitochondrial dysfunction leading to reduced oxidation under conditions of constant high supply with fatty acids fuelling triglyceride storage. We included a short comment under 3.6.

As suggested, we included other markers of lipid metabolism (CPT1A, FASN), which we did not show in the 1st draft, because no significant differences were observed. Primers are included in Table 2 now.

We did not determine triglyceride concentrations biochemically in order to avoid interference with glycogen, which is stored in high amounts in the liver. It is co-extracted to a certain degree during the chloroform/methanol extraction used in standard procedures. This makes solubility crucial, and a number of assays is interfering with reducing agents such as glucose, fructose, etc. (and glycogen). In other papers, image quantification is acknowledged for hepatic lipid content determination (e.g., NATURE COMMUNICATIONS | 8:15691 | DOI: 10.1038/ncomms15691). We think, this is acceptable, since the ductus of the manuscript relies on clearly significant differences, only, which may be reliably detected by image quantification.

Reviewer 2 Report

The authors Winkler et al are addressing the role of adipose tissue in hepatic lipid storage and further correlate the observation to the role of the immune system. Study of this crosstalk is very timely as many now report the importance of chronic low-grade inflammation to the organism homeostasis. Although the study is potentially interesting, I would recommend the rejection at this stage. 

I mention here just some major points and examples, as the point to point listing of all the problems would not be constructive at this stage. 

I find the conclusion not very well supported by the presented data. Also, many conclusions on the age-dependent increase in adipose tissues are derived from a comparison of males and females, based just on the assumption that females do not accumulate the fatty tissue (the process is slower), authors somehow like this with the innate immune system but I find this very very indirect correlation (authors are not even discussing other genetically related possible explanations). I find the experimental design weak and interpretation difficult as replicates (n) = 4 (or more but it is not specified). For example, the title is based on Figure 3b where the difference between the young and old female is ns while in the male it is significant, although the total values are obviously almost the same, all based just on image quantification. 

The poor presentation of data leads to extremely difficult navigation to described conclusions (for example Figure 8). In my opinion, the conclusions are frequently very speculative. For example the higher expression of inflammatory cytokines in used immunodeficient mice (with remaining myeloid cells - the main source of these cytokines), but I can hardly see the rationale that this is linked with adipose tissue - this is purely speculative.

Author Response

We thank this reviewer for his valuable advice and tried to meet all the concerns raised as far as possible by introducing appropriate changes in the text and the presentation of results. This improved the manuscript significantly.

Reviewer 2

The authors Winkler et al are addressing the role of adipose tissue in hepatic lipid storage and further correlate the observation to the role of the immune system. Study of this crosstalk is very timely as many now report the importance of chronic low-grade inflammation to the organism homeostasis. Although the study is potentially interesting, I would recommend the rejection at this stage. I mention here just some major points and examples, as the point to point listing of all the problems would not be constructive at this stage. 

1. I find the conclusion not very well supported by the presented data. Also, many conclusions on the age-dependent increase in adipose tissues are derived from a comparison of males and females, based just on the assumption that females do not accumulate the fatty tissue (the process is slower),

As we show in Figure 2, the size distribution of adipocytes changes to a proportion of larger adipocytes in elder as compared to younger mice, both in WT and KO mice. The increase of adipocyte size with age is well described (hypertrophy) and correlates significantly to the metabolic dysregulation and inflammatory processes, both locally and systemically, esp. in conjunction with metabolic disorders like obesity (cf. reply to comment #2). Yet, only the immune-deficient mice featured an increase in adipose tissue mass, which indicates adipose tissue hyperplasia in addition to adipocyte hypertrophy. This was most prominent in male KO mice, but also significant in female KO mice. Therefore, we agree that probably both genders featured adipose tissue remodeling. Following the reviewer´s suggestion, we tried to avoid over-interpreting of data with male mice throughout the manuscript. Concerning the comment on the rate of fat accumulation (the process is slower), we would like to emphasize that we did not measure uptake rates. We do not know whether higher amounts of fat in the livers of immune-deficient mice are due to faster accumulation as compared with the wildtype mice. It is likely, but we did not investigate the time course. Another possible mechanism is attenuation of utilization, which also leads to higher levels, but is just an accumulative effect at the same uptake rate.

2. Authors somehow like this with the innate immune system but I find this very very indirect correlation (authors are not even discussing other genetically related possible explanations).

Since principle mechanisms leading to the high incidence of fatty liver phenotypes in old individuals are described, we wanted to investigate, whether these were already initiated at younger ages without obvious signs of disease (by visual inspection of the animals). In the study presented we showed that growing older to adulthood predisposes to a fatty liver, which is aggravated by immune-deficiency. Since fatty liver diseases involve the communication from adipose tissue, we asked, whether we observe pathological changes in the adipose tissue. The answer was, yes. One potential mechanism might involve the lipidogenic capacity of the hepatocyte, which we could not confirm in our experiments with isolated hepatocytes. A second potential mechanism might involve a pro-inflammatory milieu, which we confirmed in the study presented. This milieu might derive either from the adipose tissue, but also from the innate immune system, which is known to switch from anti-inflammatory to inflammatory with increasing age. Our observations are in line with numerous recent findings showing that pathological remodeling of adipose tissue in connection with changes in the adipose secretory profile are major drivers in obesity linked with ectopic fat deposition in the liver (refs. 27-30 and 36-42). Thus, we showed that growing older drives fat storage in the liver already during adolescence, which is linked to changes in adipose tissue and potentiated by immune-deficiency. This is in line with recent reports linking aging, immunity and metabolic diseases (refs. 43,65,81) . By the way, patients with HIV infections often present with fat liver diseases without accompanying obesity, indicating that a defective adaptive immune system may promote lipid storage in the liver (cf. Inflammation and Metabolic Complications in HIV. Bourgi K, Wanjalla C, Koethe JR.Curr HIV/AIDS Rep. 2018 Oct;15(5):371-381). The mechanisms behind are similar to those we describe here.

We have tried to be more careful in our conclusions in order to avoid speculative statements, but rather pointed to the coincidence of phenotypes, which are in line with published observations.

3. I find the experimental design weak and interpretation difficult as replicates (n) = 4 (or more but it is not specified).

From our long-term experience with mice in the context of fat liver phenotypes we know that old mice display higher liver fat than young mice, even without feeding a hepatosteatosis-inducing diet. We anticipated that this is due to a process already starting earlier. Therefore, it was not the aim of this study to show differences between young and old mice. It was rather the aim to investigate the effect of growing older in the transition from adolescence to adulthood in order to show, whether already during this time period aging-related mechanisms were active to foster a fatty liver phenotype. Since we knew from our animal experiments that the immune-deficient mice featured larger fat deposits than age-matched wildtype mice, we wanted to see, whether the immune-deficient phenotype had any impact on the liver in terms of fat storage.

A minimum of 4 animals were included in the study. 4 refers to the samples used in the array experiments. In all other experiments comprising animals, a minimum of 6 animals was included. Where image analysis was performed, several slices were processed and used for quantification. We added the number of samples and animals in each section under Materials and Methods.

4. For example, the title is based on Figure 3b where the difference between the young and old female is ns while in the male it is significant, although the total values are obviously almost the same, all based just on image quantification. 

Thank you very much for this criticism, which was also raised by reviewer 1. We changed the title to a more precise description of what we have seen instead of what we concluded and avoided to mention gender. We hope that this avoids any speculative statements in the title.

Image quantification is acknowledged for hepatic lipid content determination (e.g., NATURE COMMUNICATIONS | 8:15691 | DOI: 10.1038/ncomms15691). We think, this is acceptable, since the ductus of the manuscript relies on clearly significant differences, only, which may be reliably detected by image quantification.

5. The poor presentation of data leads to extremely difficult navigation to described conclusions (for example Figure 8).

It is a little bit difficult to improve something not knowing what is bad (poor presentation?). Following also the suggestion of the Academic Editor, we changed Figures 8 to tables. But still, data are complex, which, however, may not be avoided comparing 4 groups (like in Tables 3 and 4 now), or 8 groups as in Figures 1-7. In addition, we tried to use colors and descriptions in the legends to Figures, which are largely barrier-free to ease readability for people with a handicap (e.g, bold/light in addition to red/blue). This might complicate comprehensiveness, but we would like to keep this style.

6. In my opinion, the conclusions are frequently very speculative. For example the higher expression of inflammatory cytokines in used immunodeficient mice (with remaining myeloid cells - the main source of these cytokines), but I can hardly see the rationale that this is linked with adipose tissue - this is purely speculative. 

We agree with the statement of this reviewer and have, therefore, down-toned our discussion. However, together with the group of Schalk von der Merwe (Leuven, Belgium) we have recently shown that inflammatory markers of visceral adipose tissue can accurately predict liver pathology in humans (du Plessis (2015) Gastroenterology). For instance, a decrease in SerpinE expression was among the strongest predictors for fatty liver disease. Therefore, a decrease in SerpinE in old KO mice (which is not evident in WT mice) can be discussed as one mediator in adipose tissue-liver crosstalk. Also the increase in Resistin in old KO mice could contribute to hepatic insulin resistance and increased gluconeogenesis leading to fatty liver disease in old KO mice (Steppan et al. (2001) Nature). Therefore, we believe that showing all array results of either altered or unaltered cytokines by our immune deficiency can help understanding the development of hepatosteatosis in this model, although this should be discussed more carefully. We tried to address that issue in the results section under 3.7. and in the discussion under 4.2.

Reviewer 3 Report

1. I strongly oppose calling the 28wk mice ‘old’ because they are not even middle age in human years. Jackson labs puts 28wk old mice in the human range of 30 years old. Detailing the accurate age would be preferred for each group (for example numbers like: 27.5 -/+ 2.3 weeks).

2. The title should be changed to something more accurate that highlights an accelerated deposition of fat in immune-deficient young adult mice. The way it is currently written is misleading.

3. The histology images in Figure 3a do not look representative of the bar graph results.

4. In Figure 6 the line of best fit cannot be placed over bar graphs with equal spacing for different amounts of time. For example, there are 21 hours between the first 2 data points and 48 hours between the second and third (1d-3d).  Either show bar graphs or show line graphs with time spaced appropriately.

5. Figure 8a should be shown in log scale so we can see all groups more clearly.

Author Response

We thank this reviewer for his valuable advice and tried to meet all the concerns raised as far as possible by introducing appropriate changes in the text and the presentation of results. This improved the manuscript significantly.

Reviewer 3

1. I strongly oppose calling the 28wk mice ‘old’ because they are not even middle age in human years. Jackson labs puts 28wk old mice in the human range of 30 years old. Detailing the accurate age would be preferred for each group (for example numbers like: 27.5 -/+ 2.3 weeks).

Thank you for this valuable comment. Indeed, 28 wks is not old; we agree. It was not the aim to show differences between young and old mice. From our long-term experience with mice in the context of fat liver phenotypes we know that old mice display higher liver fat than young mice. It was rather the aim to investigate the effect of growing older in the transition from adolescence to adulthood in order to show, whether already during this time period aging-related mechanisms were active fostering a fatty liver phenotype. Since we knew from our animal experiments that the immune-deficient mice featured larger fat deposits than age-matched wildtype mice, we wanted to see, whether this had any impact on the liver. In addition, we wanted to investigate, whether mechanism, which are known to be involved in the pathogenesis of fatty liver phenotypes in aged individuals, were also potentially involved at younger ages.

According to the reviewer´s suggestion, we changed the information on age in Table 1 and replaced “old” for “adult” (also in Figures and throughout the text).

2. The title should be changed to something more accurate that highlights an accelerated deposition of fat in immune-deficient young adult mice. The way it is currently written is misleading.

We do not know whether higher amounts of fat in the livers of immune-deficient mice are due to faster accumulation as compared with the wildtype mice. It is likely, but we did not investigate the time course. Another possible mechanism is attenuation of utilization, which also leads to higher levels, but is just an accumulative effect at the same uptake rate. Therefore, we changed the title to a less speculative wording to the more precise description of what we have seen instead of what we concluded and omitted to mention gender. We hope this is appropriate now.

3. The histology images in Figure 3a do not look representative of the bar graph results.

We feel that this comment may hold true only for the WTFY animals. The pictures shown were chosen randomly out of a minimum of 180 pictures. Quantification is presented with a standard deviation, which includes lower and higher fat contents. Therefore, it could easily be that the pictures do not exactly match the graph, but are in the frame of the standard deviation. We would not like to change the pictures, because that may imply manipulation.

4. In Figure 6 the line of best fit cannot be placed over bar graphs with equal spacing for different amounts of time. For example, there are 21 hours between the first 2 data points and 48 hours between the second and third (1d-3d).  Either show bar graphs or show line graphs with time spaced appropriately.

We completely agree and thank for this helpful comment. We omitted the 3h value, which should only indicate the starting point of lipid content. Since, however, differences in lipid accumulation may best be seen from the slope of the line graphs, we kept the lines.

5. Figure 8a should be shown in log scale so we can see all groups more clearly.

Also according to the Academic Editor´s comments, we used a table for presentation (now Table 3.

Round 2

Reviewer 1 Report

No more concerns.

Reviewer 2 Report

Authors have addressed most of my comments, the manuscript improved significantly. 

Reviewer 3 Report

No further comments